# Modified graph-based algorithm to analyze security threats in IoT

Ferhat Arat[1] and Sedat Akleylek[2,3,4]

[1] Department of Software Engineering, Samsun University, Samsun, Turkey
[2] Department of Computer Engineering, Ondokuz Mayis University Samsun, Samsun, Turkey
[3] University of Tartu, Tartu, Estonia
[4] Cyber Security and Information Technologies Research and Development Centre, Ondokuz Mayis University Samsun, Samsun, Turkey

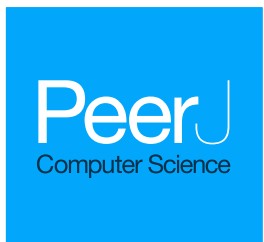

Corresponding author
Sedat Akleylek, akleylek@gmail.com

## ABSTRACT

In recent years, the growing and widespread usage of Internet of Things (IoT) systems has led to the emergence of customized structures dependent on these systems. Industrial IoT (IIoT) is a subset of IoT in terms of applications and usage areas. IIoT presents many participants in various domains, such as healthcare, transportation, agriculture, and manufacturing. Besides the daily life benefits, IIoT technology provides major contributions *via* the Industrial Control System (ICS) and intelligent systems. The convergence of IoT and IIoT systems brings some integration and interoperability problems. In IIoT systems, devices interact with each other using information technologies (IT) and network space. However, these common usages and interoperability led to some security risks. To avoid security risks and vulnerabilities, different systems and protocols have been designed and published. Various public databases and programs identify and provide some of the security threats to make it easier for system administrators' missions. However, effective and long-term security detection mechanisms are needed. In the literature, there are numerous approaches to detecting security threats in IoT-based systems. This article presents two major contributions: First, a graph-based threat detection approach for IoT-based network systems is proposed. Threat path detection is one of the most critical steps in the security of IoT-based systems. To represent vulnerabilities, a directed acyclic graph (DAG) structure is constructed using threat weights. General threats are identified using Common Vulnerabilities and Exposures (CVE). The proposed threat pathfinding algorithm uses the depth first search (DFS) idea and discovers threat paths from the root to all leaf nodes. Therefore, all possible threat paths are detected in the threat graph. Second, threat path-reducing algorithms are proposed considering the total threat weight, hop length, and hot spot thresholds. In terms of available threat pathfinding and hot spot detecting procedures, the proposed reducing algorithms provide better running times. Therefore, all possible threat paths are founded and reduced by the constructed IoT-based DAG structure. Finally, simulation results are compared, and remarkable complexity performances are obtained.

## INTRODUCTION

The rise of Internet of Things (IoT) technologies has drawn significant attention from different organizations, industries, and fields. When considering real-time benefits,

application domain contributions are indisputable. These technological developments and manufacturers will continue to be part of all aspects of life, from smart, connected systems to small appliances. Especially in industrial domains, IoT has shifted to the industrial IoT (IIoT) paradigm *via* control, management, and monitoring systems (*Jaidka, Sharma & Singh, 2020*). The IIoT systems ensure self-sufficient ecosystems in manufacturing phases *via* smart connected devices and systems such as controllers, sensors, and compact systems (*Al-Turjman & Alturjman, 2018*).

Numerous application industries, such as health care, transportation, agriculture, and security systems, are rapidly evolving *via* the form of IoT technologies (*Da Xu, He & Li, 2014*). For instance, security systems benefit from Internet-connected and addressed cameras, heat, sound, and movement sensors to provide permanent and manageable solutions (*Javaid et al., 2021*). In transportation and supply chain systems, significant data about vehicles and deliveries, such as instant location and expected delivery time, can be obtained remotely (*Wu et al., 2022*). In agriculture, there are numerous facilities and methods to provide effective farms, manufacturing, and storage. With embedded systems, sensors, and controllers, agricultural processes can be managed effectively (*Brewster et al., 2017*). Previously, the Industrial Automation and Control Systems (IACS) were largely isolated from conventional networks (*Boyes et al., 2018*). With IoT technologies and related services, Internet-connected architectures are adopted. The sub-classified system of IACS, which is termed Industrial Control System (ICS) includes any control and management systems such as Supervisory Control and Data Acquisition (SCADA) and programmable logic controllers (PLC). In these remote control structures, each data-centric sensing and transmitting device is connected *via* the Internet. An increasing number of devices and connected nodes make it easier to grow and manage all of the industrial systems (*Jaidka, Sharma & Singh, 2020*).

Industrial systems, which have become easier with innovative technologies, are always under threat due to security vulnerabilities due to the number of connected devices (*Mosteiro-Sanchez et al., 2020*). Limited device resources cause deficiencies in terms of protocols and security systems. Especially combined with the mentioned deficiencies, remote accessibility *via* Internet connection can pose a threat to the entire system. Considering all IoT system vulnerabilities, an attacker can exploit system resources and access system data. Attacks can occur with various attack types. For instance, a malicious user can access the network and cause increased network traffic (*George & Thampi, 2018*). Moreover, an attacker can monitor the network traffic. These attackers can be invisible because of device features. An attacker may access industrial control systems and cause hardware damage; therefore, the entire manufacturing and control system can be affected (*Pretorius & van Niekerk, 2020*).

Permanent, strong, and long-term security solutions are required in ICTs which are the backbone of IIoT. To identify and remove threats, various attack and vulnerability representations are designed (*Boyes et al., 2018*; *Mouratidis & Diamantopoulou, 2018a*). For instance, in *Sun et al. (2021)* and *Poolsappasit, Dewri & Ray (2012)*, graph-based representing models were designed for malware detection and risk management in IIoT systems. In *Prostov, Amfiteatrova & Butakova (2021)*, *Nandhini & Mehtre (2019)*, directed

acyclic graph (DAG)-based solutions were presented for IoT networks. DAG is a graph type that does not contain any directed cycles. In general, a directed graph structure refers to a structure consisting of edges and vertices. DAG structure does not allow for the formation of a cycle by the same vertex and edge connections. It is impossible to begin at one point in the graph and traverse the entire graph. Each edge is directed from a previous edge to the next edge. Contrary to existing intrusion detection and prevention schemes, we propose a DAG-based representation of security attacks and identify all attack paths from source to target, taking into account IIoT systems. In conventional network systems, fixed protocols and standards provide comprehensive solutions most of the time. Unlike these conventional systems, IoT and IIoT systems are quite different in terms of device features and network range. In IIoT systems, there can be a huge number of connected devices. Therefore, it is almost impossible to change the device structure and address vulnerability issues. To mitigate risk and attack scenarios, alternative and effective models are needed. The designed and proposed models should work for all attack types and scenarios.

## Motivation

In *George & Thampi (2018)*, a graph-based security framework was presented for IIoT networks. The security issues in the IIoT network were defined as related to vulnerabilities in the IIoT devices. In addition, the threat and vulnerability relations were represented as a directed graph. According to the designed graph, some attack types in devices were weighted using the Common Vulnerability Scoring System (CVSS), and threat calculations were made *via* the weighted graph. The main motivation of this article is to propose a graphical model that represents attacks on IIoT networks and depending on attacks and threats, to find attack paths on graphs and reduce these paths according to varying metrics.

There are many studies in the literature that represent attacks and vulnerabilities in IoT and IIoT network systems. Some of the related studies that are investigating and designing threat graphs were highlighted in "Related Works". In general, it is seen that various approaches were proposed to represent attacks on IoT and IIoT networks. However, the number of articles to find attack paths on IoT-based networks is very limited. In addition, very few studies focus on threat path reductions, considering threat path weights and path length. This study represents threats to the general IoT networks *via* their graph structure. Using the graph structure, each of the threats on the network can be represented as a node. In addition, the graph structure shows the relationship between the nodes clearly and with low complexity. With graph structure, all neighborhoods, weight values, and paths between the source and the destination can be shown in an organized manner. Therefore, the relationship between the threats is shown in the proposed structure as a graph. When a graph structure is generated, the DFS algorithm is used to traverse the graph and find the paths from the source to the destination. The DFS algorithm works by deeply visiting the neighbors of the visited nodes. When the visited node does not contain any child nodes, graph traversing is completed. Since the DFS algorithm works using a stack structure, it has less time and space complexity than the Breadth First Search (BFS) algorithm. Therefore, the DFS algorithm is used in the proposed procedure to find all possible threat paths. This article brings to our attention two significant advantages. First, the proposed

threat graph works regardless of the threat types that were identified by the Common Vulnerabilities and Exposures (CVE) or any other threat identification programs and databases. The second is that threat finding and reducing algorithms perform better in terms of running time, and algorithm complexity compared to *George & Thampi (2018)*. In addition, we compare our study with *Arat & Akleylek (2023a)* in terms of threat path-reducing procedure.

### Our contribution

The main contributions of the study can be summarized as follows:

- This study presents a new graph-based approach to represent general threat types in IoT-based networks.
- The graph-based approach to analyze security threats given in *George & Thampi (2018)* is modified by considering the threat detection method. This modification includes the DFS algorithm, which is traversing or searching graph algorithm, finding threat paths.
- In addition, path-reducing algorithms are merged into a compact algorithm. The modified algorithms reduce threat paths considering thresholds and compute varying values such as cumulative threat, hop length, and hot spot values on the threat path. Finally, the time complexity for hot spot detection on the threat graph is decreased by the proposed hot spot detection method.
- The proposed approach is compared to another graph-based threat assessment model given in *Arat & Akleylek (2023a)* in terms of threat path-reducing methods. According to the experimental results, the proposed idea gives better running time output in terms of threat pathfinding and hot spot detection procedures.

### Organization

The article is organized as follows: In 'Preliminaries', we give preliminaries of the study supporting graph-based approaches in the literature and some of the device threats defined in CVE and other databases. In 'The Proposed Idea', general details of the proposed approaches are presented in terms of graph structure and related procedures. In 'Experiments', proposed procedures are explained and implemented for experiments, and experimental results are highlighted. Finally, we conclude the presented study, and we give future works in 'Conclusion and Future Works'.

## PRELIMINARIES

In this section, we present the preliminaries of the study. In 'Attack graph' we give the selected attack types and their base scores using CVE and CVSS. In addition, we investigate graph-based security approaches on IoT and IIoT networks in 'Related works'. Table 1 illustrates abbreviations and variables used throughout the article.

### Attack graph

In this subsection, some IoT-based vulnerabilities are presented. In our assumptions, these vulnerabilities are used to generate a threat graph. There are many presented

**Table 1 Abbreviations and variables used through the article.**

| Variable | Definition |
|---|---|
| TDG | IoT-based threat directed graph |
| src | Source node |
| dest | Destination node |
| graph | Generated threat graph |
| visited | Visited node in pathfinding procedure |
| path | Each path for src and dest pairs |
| pathList | List of all paths in Algorithm 1 |
| hop | Number of hop count |
| listOfPath | Dimensional list of all paths in Algorithm 2 |
| threatThreshold | Threat threshold |
| remainingPath | Path list to store remaining paths after path reduction |
| removedPath | Path list to store removed path after path reduction |
| threatSum | Cumulative threat weight |
| listOfPathThreat | Dimensional list of all threats in Algorithm 3 |
| hopThreshold | Hop length threshold |
| adjDict | Hash object to store node and threat pairs |
| hotSpotDict | Hash object to store node, threat, and hot spot |
| alpha | Hot spot threshold |

vulnerabilities in IoT-based networks. These attacks and threats are defined and classified in various databases. CVE is a database that identifies and classifies cyber security vulnerabilities (*CVE, 2023*). Various vulnerabilities and threats are discovered by organizations, and CVE publishes these vulnerabilities. In this work, we define some of these vulnerabilities, which are related to our assumptions. For example, CVE-2014-2360 is a vulnerability that allows malicious users to execute arbitrary codes. It exists in wireless sensor I/O modules. CVE-2022-25359 is another vulnerability that exists in SCADA controllers. It allows unauthenticated users to change system files. CVE-2022-0162 exists in some wireless router types, and an attacker may intercept router credentials and perform management operations *via* the wireless router interface. CVE-2011-2688 is a SQL injection vulnerability that exists on web servers. Through CVE-2019-2776, an attacker may cause intrusions into the Oracle database. CVE-2021-4045 vulnerability exists on IP cameras and allows attackers to control all camera activities. CVE-2021-30353 vulnerability may cause improper validation in Snapdragon IIoT. Table 2 shows some published vulnerabilities and their Version 2 Base Scores (BS) in the CVE program and the National Vulnerability Database (NVD), which exist in IoT-based systems.

## Related works

In this subsection, we investigate proposed security models. There are several works for IIoT and IoT systems. Therefore, we consider graph-based intrusion and threat detection models in the literature in the investigation. The main differences between the literature

**Table 2 Some of IoT-based vulnerabilities.**

| CVE-ID | BS (v2.0) | Description |
|---|---|---|
| CVE-2020-3162 | 5.0 | CoAP vulnerability |
| CVE-2021-30353 | 5.0 | Qualcomm Snapdragon IIoT vulnerability |
| CVE-2019-2776 | 5.5 | Oracle Database vulnerability |
| CVE-2021-22779 | 6.4 | PLC vulnerability |
| CVE-2014-2360 | 7.5 | Wireless sensor vulnerability |
| CVE-2011-2688 | 7.5 | Web Server vulnerability |
| CVE-2022-25359 | 9.1 | SCADA controller vulnerability |
| CVE-2022-0162 | 9.8 | Wireless router vulnerability |
| CVE-2021-4045 | 10.0 | IP Camera vulnerability |

and our study are we consider graph-based threat modeling approaches to present an approach. We also use algorithmic methods to detect, compute, and reduce risks, and we improve the literature in terms of complexity and running time using varying existing and customized algorithms. In addition, we propose a generic vulnerability assessment method to apply for each platform. Table 3 summarizes graph-based security approaches in the literature.

In *Wang et al. (2018)*, the vulnerability assessment model was designed in IIoT. The designed model works graph-based and considers the maximum flows of the path. In an attack graph, nodes represent the host property, and edges represent attacks or vulnerabilities. The method considers node and edge relationships to identify maximum flows. The attack risk was calculated by graph weight, and the maximum loss flow represents the attack path. In addition, evaluation results were demonstrated under various nodes.

In *Qureshi et al. (2020)*, a routing protocol for low-power and lossy networks (RPL)-based threat detection model was proposed. The proposed model performs by detecting the different types of attacks, such as Sinkhole attacks, HELLO-Flood attacks, and Blackhole attacks, in an IIoT environment. With the rank structure of RPL messages, the message transmitting time was considered as a determining factor of malicious nodes. The idea relied on threshold values. The experimental results were compared by a standard RPL algorithm.

In *Sun et al. (2021)*, a malware detection scheme was proposed. The proposed scheme considers malware behavior graphs. Graph optimizations and malware classification were made on malware types such as Delf, Obfuscated, *etc.* The experimental results were evaluated in terms of true positive rate accuracy.

In *Nguyen et al. (2022)*, a supervised machine learning classification method was proposed. A graph-based hybrid analysis method, which contains static and dynamic methods, was used in IoT botnet detection. Malware classification was made using printable string information graph (PSI) features *via* machine learning-based algorithms such as decision tree (DT) and k-nearest neighbor (kNN).

**Table 3 Highlights of the related works.**

| Ref. | Year | Description | Complexity | Platform | Advantages/Disadvantages |
|---|---|---|---|---|---|
| *Poolsappasit, Dewri & Ray (2012)* | 2012 | Proposed Bayesian attack graph to security risk assessment. A method developed to estimate an organization's risk level. Edges represent posterior probabilities, and vertices represent various device and attack attributes. | The graph generation is $O(n^2)$, Computational complexity of prior or posterior cases is $O(2^n)$. | General network system | Proposed model is not scalable for large networks. No method was proposed to reduce the attack path. |
| *Wang et al. (2018)* | 2018 | Finds maximum loss flow in graph. Nodes represent hop and edges represent threats. Works on a directed graph. | Vulnerability detection is $O(n.p)$, Maximum loss flow is $O(n)$, Path seeking is $O(n^2)$. Total time complexity is $O(n^2)$. | IIoT | No methods were proposed to reduce attack paths or edges. |
| *George & Thampi (2018)* | 2018 | Finds and reduces attack paths in graph. Vertices represent the vulnerabilities, edges represent relations between vulnerabilities. Works on a directed graph. | Threat path detection is $O(N_{ij}.L_{ij})$, Threat and hop length calculation is $O(N_{ij})$, Path reduction is $O(N_{ij}L_{ij}pq)$ Hot spot detection is $O(n^2)$. | IIoT | Final results were calculated manually. The graph was designed for IIoT networks. In generating subgraphs, separated procedures were used. |
| *Polatidis, Pavlidis & Mouratidis (2018)* | 2018 | Proposed attack path identification approach. Attack graph approach was used. Attack paths were identified using the DFS algorithm. | The complexity of algorithms were not analyzed. | General network system | Real time data was used. Threat weights were not considered. |
| *Mouratidis & Diamantopoulou (2018b)* | 2018 | Proposed security analysis method for IIoT. The proposed method includes two phases. Each IIoT component was represented as an actor. | The complexity of algorithms were not analyzed. | IIoT | Real time data was used. Potential attack paths were identified and categorized considering importance. Attack paths were not filtered, considering their importance. |
| *Qureshi et al. (2020)* | 2020 | Detects different type of attacks using RPL. Genetic algorithm was used to detect presence of security threats. Nodes represent devices. Works on DODAG. | The complexity of algorithms were not analyzed. | Smart home IIoT | Proposed algorithms work only detects vulnerabilities different type of attacks. Threat weights were not considered. |
| *Sun et al. (2021)* | 2021 | Proposed graph-based malware detection architecture. Malware detection scheme works based on classification. | The complexity of algorithms were not analyzed. | IIoT | Proposed methods only detect malware considering behavior graph. Threats and path weights were not considered. |
| *Stellios, Kotzanikolaou & Grigoriadis (2021)* | 2021 | Proposed graph-based attack path identification and risk assessment method. Interaction modeling was performed to define device relations. | For the graph construction phase, computational cost is $O(D^n)$ where $D$ is set of all the devices, and $n$ is the number of interactions. | IoT | Proposed methods works using pre-defined interactions. Some devices can interacts with each others without pre-defined connections. |
| *Sukiasyan et al. (2022)* | 2022 | Proposed security architecture to mitigate security risks. The model works based on blockchain DAG structure. Different types of attack vectors such as tampering, and DDoS were analyzed, and security data exchange was ensured. Works on DAG. | The complexity of methods were not analyzed. | IIoT | Threat weights were not considered. Only communication paths were considered. |

(Continued)

| Table 3 (continued) | | | | | |
| --- | --- | --- | --- | --- | --- |
| Ref. | Year | Description | Complexity | Platform | Advantages/Disadvantages |
| *Nguyen et al. (2022)* | 2022 | Presented a novel to detect IoT botnets using PSI-subgraph features with machine learning-based algorithms. Malware classification was made and traversed nodes were removed to reduce algorithm implementation time. Works on a directed graph. | The complexity of algorithms were not analyzed. | IoT/IIoT | Proposed methods only classify malware as botnet or benign. No methods were proposed to detect attack paths. |
| *Jing & Wang (2022)* | 2022 | Proposed graph theory based DDoS attack classification method. Machine learning approach was used to detect attacks. Vertices represent IP addresses and ports, and edges represent relationships between vertices. Works on directed graphs. | The complexity of algorithms were not analyzed. | IoT | Proposed method classify DDoS attacks. Adjacency matrix was used to identify relationships. No methods were proposed to detect the attack path. The number of partitions was calculated manually. |
| *Arat & Akleylek (2023a)* | 2023 | Proposed graph based threat assessment model. DFS and Floyd-Warshall algorithms were used in path finding and threat computing. Vertices and edges represent nodes and relations, respectively. Works on directed graphs. | The time complexity of pathfinding algorithm is $\mathcal{O}(n^2)$ The time complexity of threat computing algorithm is $\mathcal{O}(n^3)$. The time complexity of path filtering algorithm is $\mathcal{O}(n^3)$. | General networks system | Threat computing procedure takes $\mathcal{O}(n^3)$ time. Path filtering procedure depends on threat computing and it takes $\mathcal{O}(n^3)$ time. Hop count and threat amount were used as filtering metric. |
| *Arat & Akleylek (2023b)* | 2023 | Proposed graph-based risk assessment model. Vulnerable paths were detected. Risk computing equations were presented. Works on DAG. | Path detection is $O(V + E)$. Threat computations procedures work $O(n)$, Path filtering procedures works $O(n)$. | General network system | Running time for path detection is reduced. Risk levels were determined overall network. A complete risk assessment model was presented. |
| Ours | 2023 | Proposed graph-based threat detection model. Threat pathfinding procedure is modified. Vertices represent threats, and edges represent threat weight. Works on DAG. | Path detection is $O(V \times V)$, Threat and hop length computation is $O(N_{ij}.L_{ij})$, Path reducing is $O(P_{ij})$, Hot spot detection is $O(V)$. | General network system | Running time for path detection is reduced. Hot spot detection complexity is reduced. The final results are calculated inside the procedures. |

In *Jing & Wang (2022)*, a graph theory-based DDoS attack detection method was proposed for IP ports from source to destination. Edge and vertex structures were generated to extract traffic data characteristics. Next, clustering and classifying were done *via* principal component analysis (PCA) and Fuzzy C-means (FCM) clustering methods. The proposed model works in IP-connected topologies, considering traffic flows.

In *Poolsappasit, Dewri & Ray (2012)*, a Bayesian network-based risk assessment framework was proposed. The designed model incorporates different relationship models such as attack graphs and attack trees. The initial test vulnerabilities were generated using CVSS, and network attacks were modeled as a Bayesian attack graph (BAG). To manage network risks, the graph was generated considering the probabilities of attributes. With the

BAG structure, pre-conditions and post-conditions were highlighted in terms of vulnerability exploitation.

## THE PROPOSED IDEA

In this section, we present the details of the proposed idea. We also explain the designed and modified algorithms *via* separated subsections. The main differences from *George & Thampi (2018)* are vulnerability path detection methods and path reduction procedures. In *George & Thampi (2018)*, the authors used a recursive pathfinding algorithm. In our approach, the graph searching and traversing method, which is known as DFS, is used to detect vulnerabilities in graphs. In addition, subgraph-generating algorithms are merged without any additional procedures.

An attack graph is a network security application of graph theory that can be used to detect an attack or vulnerability route, with the node representing the hosts' state or properties (*Szwed & Skrzyński, 2014*). In this definition, an attack or vulnerability graph can be represented according to host and attack types, pre or post-threat conditions, and host and attack relations, considering attack weights. Therefore, possible threat paths can be detected using the graph theory. For this reason, this article uses attack attributes and weights representing general IoT-based network topology to analyze threat paths and network vulnerabilities.

In Fig. 1, the proposed threat detection model is demonstrated with three basic tasks. The main tasks of the proposed model include the following tasks:

- Considering IoT-based device connections, network elements are represented as a vulnerability or threat. Since each network element has a vulnerability defined in CVE or other databases. To generate graph topology, nodes, and connection links are connected in a DAG structure.
- Using the CVSS scores, communication links are weighted, and vertices are referred to as attack types. According to the generated graph structure, all possible attack paths from source to destination are detected using a DFS-based discovery algorithm.
- Considering attack paths, weighted links and the number of paths are reduced using hop count, threat threshold, and hot spot threshold values.

### Representation of threat graph

In this section, we give the general threat graph representation notations. During the graph construction phase, we assume that the configured attack graph works independently from threats or vulnerabilities. In other words, the designed graph works similarly when the attack or vulnerability characteristics are changed. We construct a general graph model to define our proposed procedures. It can be a conventional network or an IoT-based network. In the representation of TDG, we assume that each node in the graph represents an IoT-based network device such as a sensor, router, or IP-based component. As

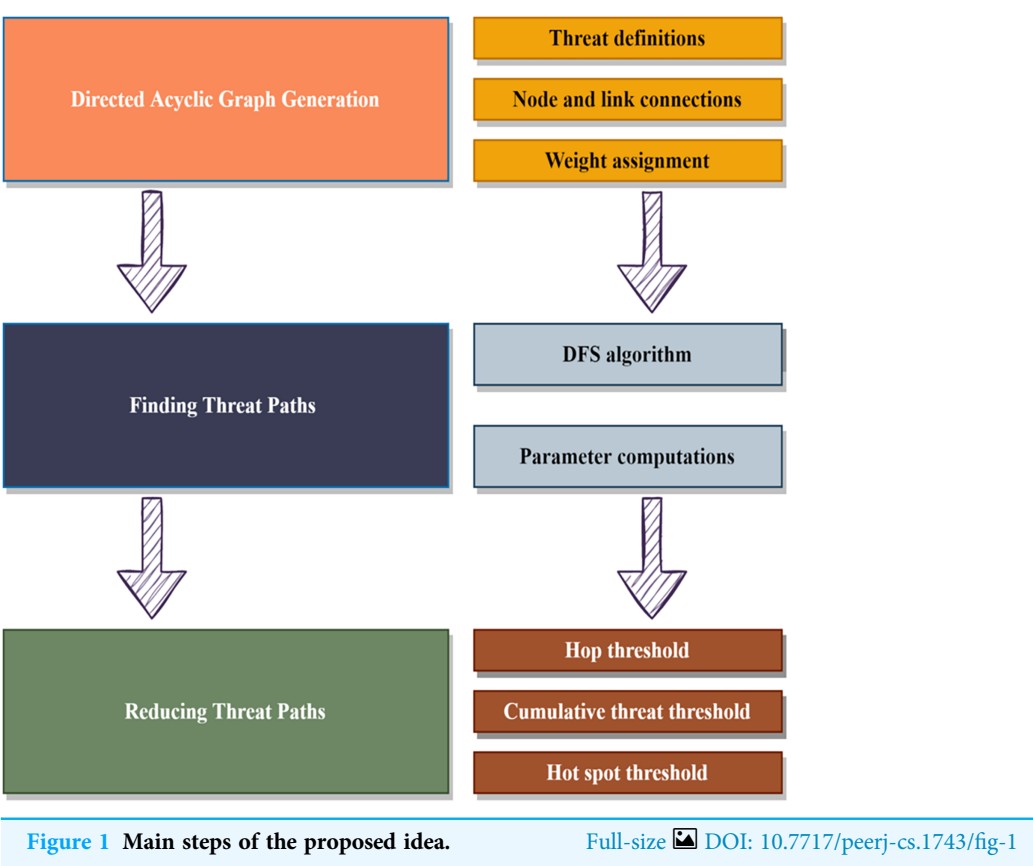

**Figure 1 Main steps of the proposed idea.**

explained in 'Attack graph', these devices have vulnerabilities that have the potential to affect other devices and systems. In addition, IoT-based network devices have predefined CVSS scores in CVE to determine vulnerability levels. We assign these CVSS scores to edges. In general, each edge is a connection between vertices. Firstly, a device affects a connected device or path depending on its vulnerability. CVSS values consist of base scores provided by the National Vulnerability Database (NVD). These scores represent the unique characteristics of each vulnerability. Since the weighted graph structure is used, we created the weight assignments to the edges in the graph by directly considering the CVSS values between 0 and 10, to be simple and understandable. Due to a representative threat value calculation being used, a different calculation metric or formulation is not used. We focused on calculating the threat value and reducing it according to the threshold value. Therefore, we assign these scores to the links to generate a weighted graph. Then, we compute the path risk level after identifying a communication path using CVSS scores. First, we identify the exploitable threats and elements in the IoT-based network. We define the IoT-based Threat Directed Graph (TDG) formally as follows:

$$TDG = (N, src, dest, T_t, \lambda, E_w)$$

1) $(V, E)$ is a directed acyclic graph, which consists of vertices and edges.

2) $N$ represents the set of nodes in TDG. As the algorithm parameter, each node stores a threat in an IoT-based network.

3) *src* is the source node that initializes the attack path. It acts as a root node in a graph.

4) *dest* is the destination node that directly acts as a sink node in a graph. It can be named as a target node.

5) $T_t$ represents the set of threats in a graph.

6) $\lambda$ is the threat weight, which is defined in CVSS.

7) $E_w$ represents the link of the graph. It is constructed considering CVSS scores and stores $\lambda$ values between two nodes in a graph.

We considered some general network types to realize our graph. In our scenario, there are nodes and their interactions. We defined these pairs as vertices and edges $(V, E)$. In our assumptions, nodes represent the IoT-based devices that have vulnerabilities and corresponding scores. We also have some edges, which represent dependencies between nodes and vulnerabilities. These dependencies are randomly generated by the custom Python simulator. After the network graph is generated, it can be paired with any device and link, such as an IP-based sensor and network router. For instance, we explain it using a case scenario as shown in Fig. 2.

Figures 3A and 3B illustrates an example representation for TDG. According to the figure, *src* is the source node, and *dest* is the destination node. In addition, communication link values between nodes are represented by $\lambda$ which is defined as a threat weight. Additionally, Figs. 4–6 in the following sections, show the change of the main graph (Fig. 3B) as a result of the parameters included in the TDG definition and the proposed algorithms. Thus, the current graph structure that is updated due to the performing of the relevant algorithms is visualized.

## Detecting of all threat paths

In this subsection, a threat pathfinding algorithm is proposed. A DFS-based traversing graph algorithm is used to find all possible attack paths from *source* to *destination*. There can be many paths in a directed graph. Finding all paths that will be used by attackers while calculating the total threat amount is a very heavy load for system administrators. Since detected attacks can be prevented in this way, Algorithm 1 has seven inputs. The inputs are as follows, respectively:

- IoT-based network graph *TDG*,
- Source node *src*,
- Destination node *dest*,
- Boolean value to determine visited nodes,
- List of detected attack nodes,
- List of all detected attack path,
- Hop count parameter.

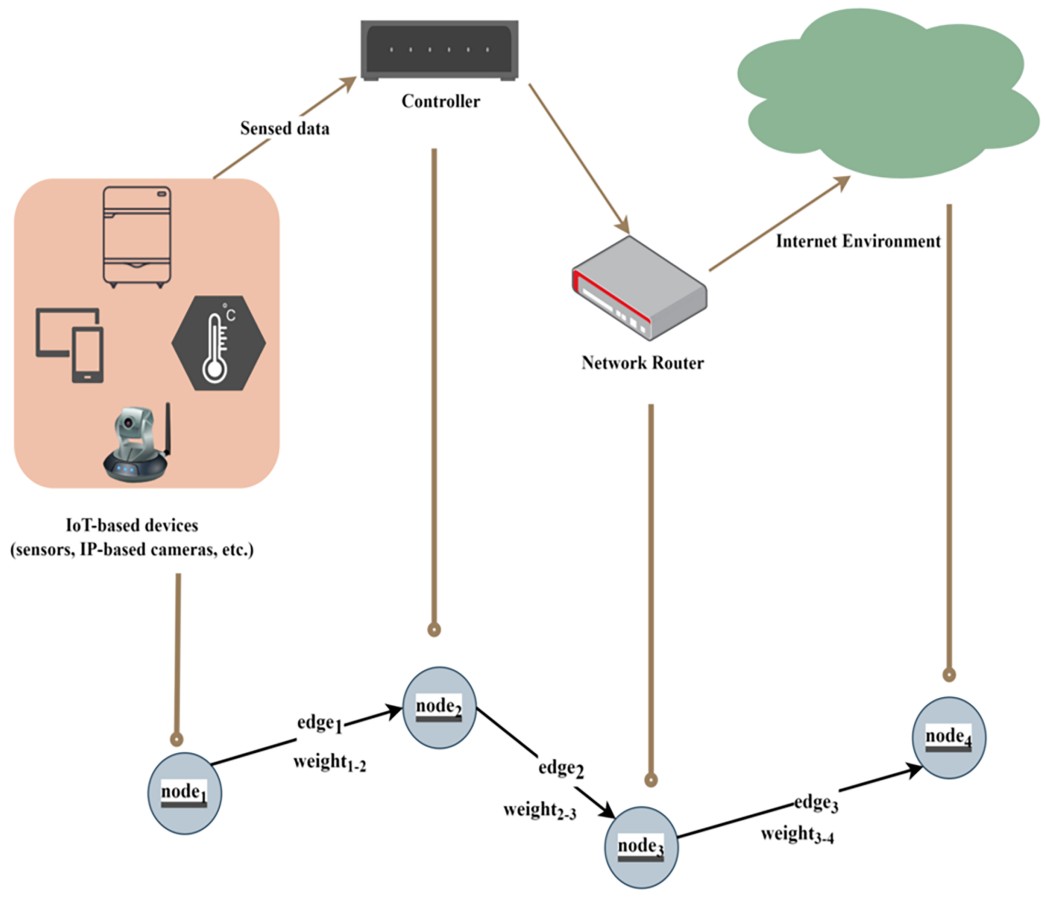

**Figure 2  Generalized device-based case scenario.**

Algorithm 1 demonstrates the main steps of threat path detection. To find possible threat paths from *src* to *dest* in the given as a parameter TDG graph, GETALLPATHSBYDFS (graph, src, dest, visited, path, pathList, hop) is called. The proposed method works using the DFS algorithm, and the complexity depends on the total number of vertices.

When all possible threat paths are found *via* Algorithm 1, the total amount of number-of-hop and threats are calculated *via* Algorithm 2. The algorithm takes two parameters. In the given *graph*, the number-of-hop is calculated according to the table *listOfPath*. Since the number of path values is not stored in a parameter, it is calculated considering the length of each path list in the table *listOfPath*. The algorithm time complexity is $\mathcal{O}(P_{ij})$ where the $P_{ij}$ is the number of the path between source $i$ and destination $j$. Since the algorithm runs for each path in the *listOfPath*.

## Path reduction with threat threshold

In this subsection, we give the path reduction procedure, which works based on threat threshold metrics. A threat path with a high threat amount is more desirable for an attacker in IoT-based TAG. In addition, it will be easier for system administrators to focus on critical nodes and points with high threat value instead of nodes with low threat value in terms of ensuring system security and reliability. Therefore, reducing attack paths with

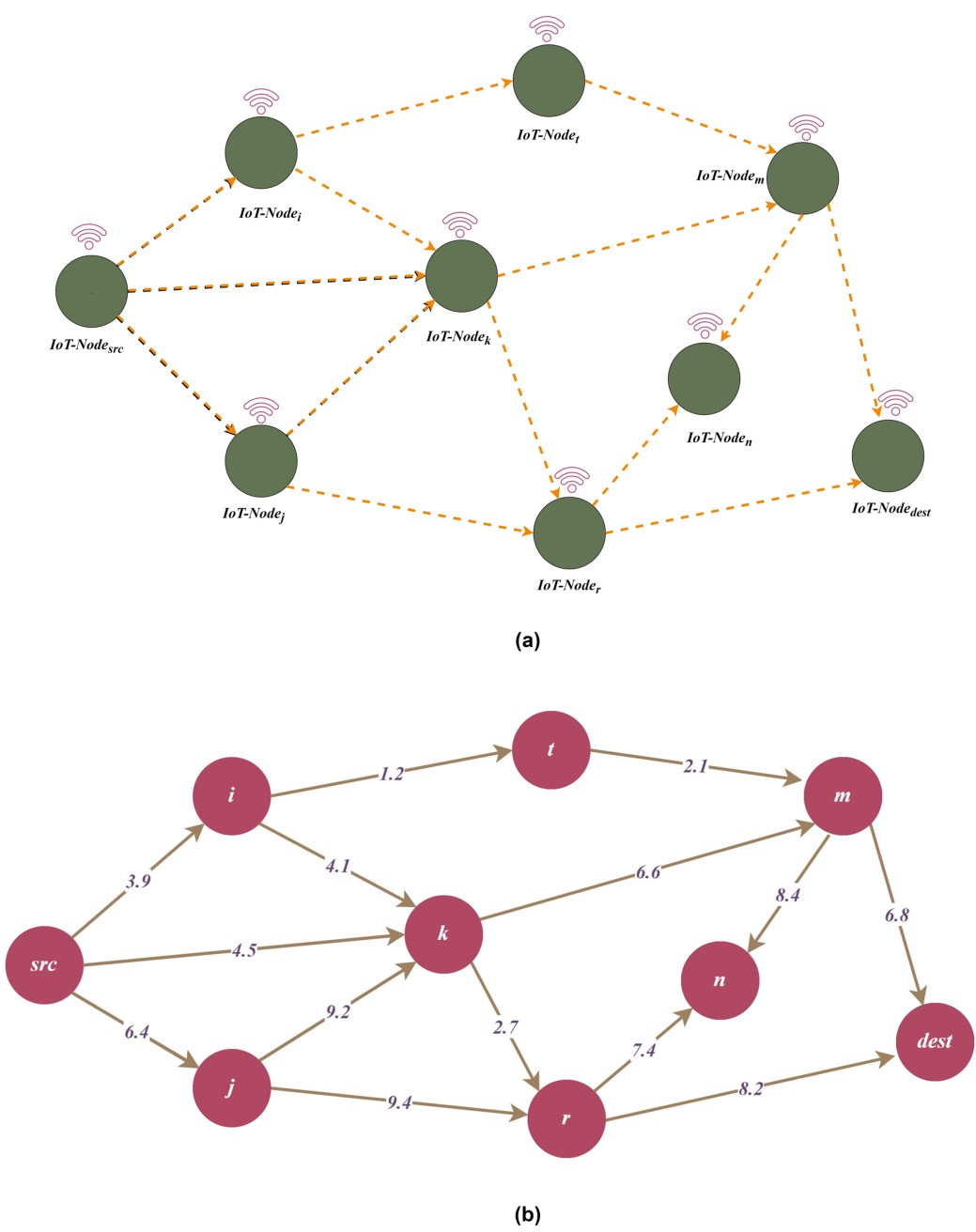

**Figure 3 (A)** A generic representation for threat directed graph (TDG). **(B)** A weighted example for threat directed graph (TDG) representation.

high total threats is essential for security mechanisms. To reduce these types of attack paths, a threat threshold value is used.

Algorithm 3 demonstrates a threat path-reducing procedure. The algorithm takes three inputs. *listOfPathThreat* table stores paths and their total threat values. The algorithm gives a subgraph of $TDG = (V, E)$ which includes removed and remaining paths. The remaining path list is generated considering if the total threat of the path is greater than the

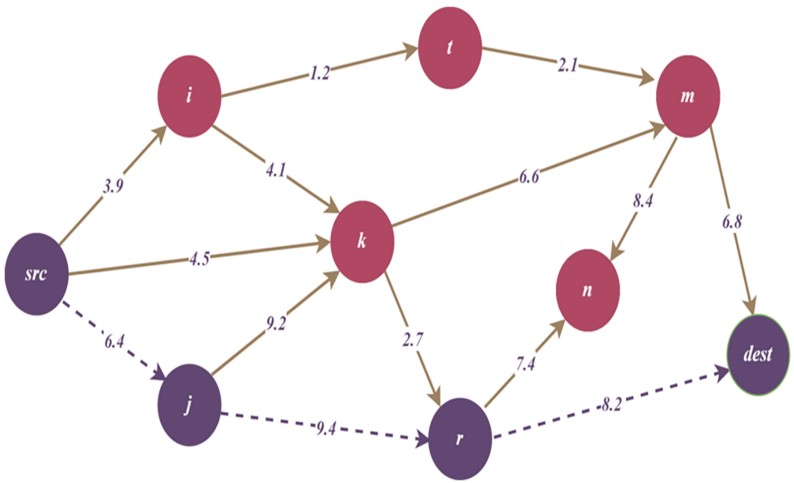

**Figure 4 Remaining TDG after reducing path by cumulative threat threshold.**

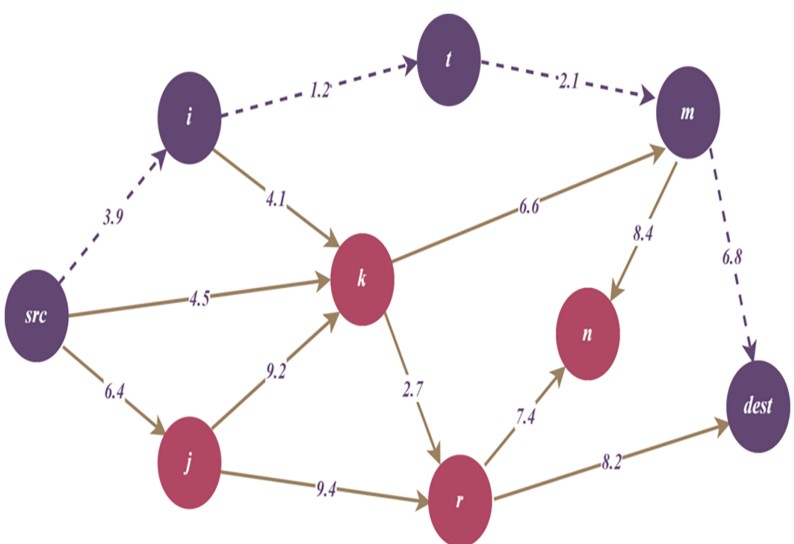

**Figure 5 Remaining TDG after reducing threat path by hop length threshold.**

threat threshold *threatThreshold* value. The time complexity of the procedure is given by $\mathcal{O}(P_{ij})$ where $P_{ij}$ is the number of paths from source $i$ to destination $j$.

Figure 4 illustrates the remaining TDG after reducing the path by maximum cumulative threat threshold, and dashed lines represent reduced path $\{src - j - r - dest\}$.

## Path reduction with hop length

In this subsection, we give a path reduction procedure that works based on hop length threshold metrics. In vulnerable network topology, more hop counts state is not desired by an attacker since increasing the number of hop counts increases the attack detecting probability. When an attacker wants to reach a destination or target device, it needs to pass through more security mechanisms or a detection strategy. Additionally, considering the

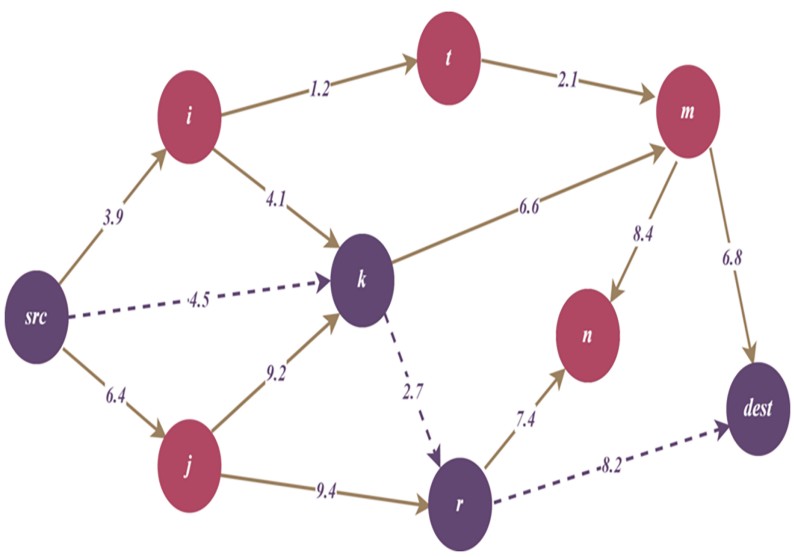

**Figure 6 Remaining TDG after path reduction by hot spot threshold.**

---

**Algorithm 1 Algorithm to find possible attack paths.**

**Input:** A TDG *graph*, source node *src*, destination node *dest*, visited boolean variable *visited*, path lists, and hop length variable *hop*.

**Output** A table *listOfPath* containing the threat paths from *src* to *dest*.

1: **Procedure** GETALLPATHSBYDFS(graph, src, dest, visited, path, pathList, hop)
2:      *visited*[*src*] ← *True*
3:      Add src to table *path*
4:      *hop* + +
5:      **if** *src* = *dest* **then**
6:          Compute the threat for the *path*
7:          Add path to table *pathList*
8:      **else**
9:          **for each** *node* adjacent to *src* **do**
10:              **if** *visited*[*node*] ← *True* **then**
11:                  GETALLPATHSBYDFS(graph, node, dest, visited, path, pathList, hop)
12:              **end if**
13:          **end for**
14:      **end if**
15:      Pop to last inserted node from table *path*
16:      *path.pop()*
17:      *visited*[*src*] ← *False*
18: **end procedure**

---

---

**Algorithm 2** Algorithm to compute hop and threat values.

**Input:** A TDG *graph*, path list *listOfPath*.

**Output** A set of threat and hop length parameters.

1: **Procedure** FINDVALUES (graph, listOfPath)

2:         *totalHop, avgHop, totalThreat* ← 0

3:         *minHop* ← ∞

4:         *maxHop* ← −∞

5:         *minThreat* ← ∞

6:         *maxThreat* ← −∞

7:         **for each** *path* **in** *listOfPath* **do**

8:           **if** *minHop* > length of ((*path*) − 1) **then**

9:             *minHop* ←length of ((*path*) − 1)

10:           **end if**

11:           **if** *maxHop* < length of ((*path*) − 1) **then**

12:             *maxHop* ←length of ((*path*) − 1)

13:           **end if**

14:           *totalHop+* = (length of ((*path*) − 1)

15:         **end for**

16:         **for each** *thr* **in** *listOfPath.thrList* **do**

17:           **if** *minThreat* > *thr* **then**

18:             *minThreat* ← *thr*

19:           **end if**

20:           **if** *maxThreat* < *thr* **then**

21:             *maxThreat* ← *thr*

22:           **end if**

23:           *totalThreat+* = *thr*

24:         **end for**

25: **end procedure**

---

overall system, a higher number of hops means more time and protection mechanisms. In this way, it will be difficult for malicious users or threats to damage the device and the network. Therefore, threat paths with a high number of hops are reduced in common conventional threat-based mechanisms.

Algorithm 4 demonstrates a path path-reducing procedure which considers the number of hop counts. The algorithm performs using three inputs and produces a subgraph of $(V, E)$ which includes removed and remaining paths. The hop threshold determines the maximum desired path length in *pathList*. Finally, REDUCEPATHBYHOP(graph, listOfPath, hopThreshold) procedure consists of two tables as a output. The time complexity of the algorithm is given by $\mathcal{O}(P_{ij})$ where $P_{ij}$ is the number of paths from source $i$ to destination $j$.

---

**Algorithm 3** Algorithm to reduce paths by threat.

**Input:** A TDG *graph*, path lists *listOfPath*, and threat threshold value *threatThreshold*

**Output** Tables *remainingPath* and *removedPath* containing removed and remaining paths.

1: **Procedure** REDUCEPATHBYHOP (graph, listOfPath, threatThreshold)

2:    *listOfPathThreat* object contains all paths and theirs sum of threat as last element of array.

3:    *remainingPaths* ← Ø

4:    *removedPaths* ← Ø

5:    **for each** *threatSum* **in** *listOfPathThreat*[−1] **do**

6:        **if** *threatSum > threatThreshold* **then**

7:            *remainingPaths* ← *listOfPathThreat*[0]

8:        **else**

9:            *removedPaths* ← *listOfPathThreat*[0]

10:        **end if**

11:    **end for**

12: **end procedure**

---

**Algorithm 4** Algorithm to reduce paths by hop.

**Input:** A TDG *graph*, path lists *listOfPath*, and hop threshold value *hopThreshold*

**Output** Tables *remainingPath* and *removedPath* containing removed and remaining paths.

1: **Procedure** REDUCEPATHBYHOP (graph, listOfPath, hopThreshold)

2:    *remainingPaths* ← Ø

3:    *removedPaths* ← Ø

4:    **for each** *path* **in** *listOfPath* **do**

5:        **if** length of $(path) - 1 > hopThreshold$ **then**

6:            *removedPaths* ← *path*

7:        **else**

8:            *remainingPaths* ← *path*

9:        **end if**

10:    **end for**

11: **end procedure**

---

Figure 5 illustrates the remaining TDG after reducing the path by hop length threshold, and dashed lines represent reduced path $\{src - i - t - m - dest\}$.

## Detection of hot-spots

In this subsection, we give the hot spot detection procedure, which works based on hot spot threshold metrics. High relations between network devices with each other can be considered a vulnerability in traditional and IoT-based networks. Connections can pose a

---

**Algorithm 5** Algorithm to find hot spots.

**Input:** A TDG *graph*, hash object *adjDict*, and hot spot threshold value *alpha*.

**Output** A hash object *hotSpotDict* containing nodes and hot spots.

1: **Procedure** FINDHOTSPOTS(graph, adjDict, alpha)

2:     *adjDict* object contains all nodes and their connections as key and value pairs.

3:     $hotSpotDict \leftarrow \{\}$

4:     $hotSpotDict \leftarrow adjDict$

5:     **for each** *node* **in** *adjDict.keys()* **do**

6:         $hotSpotDict[node] \leftarrow [node, threatLabel, alpha]$

7:     **end for**

8:     **return** *hotSpotDict*

9: **end procedure**

---

threat to devices. For example, the IoT-based device, which has many connections as input and output, acts as a HUB device, which provides many uncontrolled connections. Therefore, heavy connected node loads must be detected and reduced. In Algorithm 5, we propose a procedure to detect nodes that have a high connection.

The procedure *FINDHotspots (graph, adjDict, alpha)* performs using *alpha* hot-spot threshold value and returns a hash object. The hash object ensures an efficient, convenient structure to quick data storage and retrieval. The hash object stores and retrieves data based on lookup keys. Therefore, it provides time efficiency and prevents data duplication. The hash object *hotSpotDict* contains key and value pairs that are formed by nodes and hot-spot degrees. In this algorithm, key and value pairs represent nodes and their individual hot spot degrees. In this way, we access the high and low connected nodes and paths quickly due to the average time complexity of the hash data structure in $\mathcal{O}(1)$ time. High-connected IoT-based network nodes can be reduced using the returned hash object. The time complexity of the algorithm is given by $\mathcal{O}(V)$ where $V$ is the number of vertices.

Figure 6 illustrates the remaining TDG after path reduction by hot spot threshold, and dashed lines represent reduced path $\{src - k - r - dest\}$. As seen in the figure, node $k$ has the highest number of connections; therefore, node $k$ can be considered as a hot spot maximum node.

## Comparison

In this subsection, we compare our study with *George & Thampi (2018)* in terms of algorithmic approaches and complexity. According to the comparison, our modified threat-finding algorithm works based on DFS. Therefore, it works recursively. Also, in Algorithm 2, the hop length variable is generated using the length of the path in the list. In addition, remaining and reduced path lists are generated using a single and compact procedure. In *George & Thampi (2018)*, pathfinding algorithm complexity is related to the number of paths and path length. In our approach, the time complexity depends on the

**Table 4 Comparison of proposed algorithms.**

| | Proposed algorithm | *George & Thampi (2018)* |
|---|---|---|
| Finding all paths | The algorithm performs recursively, and the content of the path list decreases for each loop because of the stack object. The time complexity is $\mathcal{O}(V \times V)$ where $V$ is the number of vertices. The auxiliary space is $\mathcal{O}(V \times V)$ where $V$ is the number of vertices. Parameters were stored by a single path list. | The algorithm performs recursively, and the content of the path list increases for each loop. The time complexity is $\mathcal{O}(N_{ij}.L_{ij})$ where $N_{ij}$ is the number of paths and $L_{ij}$ is the hop-length of the longest path. General table $T$ was used to store all parameters. |
| Calculating hop and threat values | The algorithm performs linear. The time complexity is $\mathcal{O}(P_{ij})$ where $P_{ij}$ is the number paths. The hop length was generated using path length. | The algorithm performs linear. The time complexity is $\mathcal{O}(N_{ij})$ where $N_{ij}$ is the number of paths. The hop length variable was generated using by different variable. |
| Reducing paths by hop and threat | The algorithm performs linear. The time complexity is $\mathcal{O}(P_{ij})$ where $P_{ij}$ is the number paths. Remaining and reduced path lists are generated *via* compact and collective single function. Reduced subgraph is generated in a single function. | The algorithm performs linear. The time complexity is $\mathcal{O}(N_{ij}L_{ij}pq)$ where $N_{ij}$, $L_{ij}$, p and q represents the number of paths, The hop-length of longest path among them, the number of attackers, and the number of targets respectively. Remaining and reduced path lists were generated *via* separated algorithms. Reduced subgraph was generated *via* separated algorithms. |
| Finding hotspots | The algorithm performs linear. The time complexity is $\mathcal{O}(V)$ where $V$ is the number of vertices. Hash data type is used to store labels and the hot-spot index of the nodes. | The algorithm performs quadratic. The time complexity is $\mathcal{O}(n^2)$ where $n$ is the number of nodes. Adjacency matrix was used to store labels and hot-spot index of the nodes. |

number of vertices. In *George & Thampi (2018)*, attack pathfinding algorithm complexity was given as $\mathcal{O}(N_{ij}L_{ij})$ where $N_{ij}$ is the number of paths and $L_{ij}$ is the hop-length of the longest path. In the modified approach, the algorithm complexity is $\mathcal{O}(V \times V)$ where $V$ is the number of vertices. In the hot spot finding procedure, the algorithm complexity is reduced to $\mathcal{O}(V)$ where $V$ is the number of vertices. In summary, we improved previous work in different aspects. As mentioned, we have made improvements in terms of complexity and running time. The most obvious of these improvements are the path-finding algorithm used, the calculation of the threat value within the same algorithm, the use of data structures that provide access and storage efficiency, and the graph reduction phase in a single procedure. Table 4 highlights general differences between studies.

# EXPERIMENTS

## Experimental environment

In this section, we present the setup of the simulation environment. We design a custom simulator using the Python programming language. Thus, the simulation is operated on the custom simulator. A threat-based graph, which consists of fixed-located nodes with IoT-based network devices, is generated. Each link is weighted using pre-defined threat types. In the simulation phase, it is assumed that the root node, which acts as an attacker, targets to reach the sink node by passing through nodes. Figure 7 illustrates an instance of

an IoT-based network threat graph. The graph topology is designed to support all threat types. In other words, the proposed algorithms run considering general threat and attack types that were defined in CVE. The simulation is performed on the same machine, which has an Intel Core i5 10300H CPU, 16 GB of RAM, and a GTX 1650 GPU.

## Example scenario

In this subsection, we present an example IoT-based threat graph scenario. In this scenario, a threat graph was generated using nine threat nodes. Node connections and locations were randomly distributed. In addition, the threat weights were set using CVSS metrics. In general, we propose an attack or threat path detection approach using graph theory. We use the DFS algorithm to traverse the graph and identify all paths from source to destination. In addition, we propose threat path-reducing procedures to assess and mitigate risk in the IoT network topology to provide a more secure communication structure. Tables 5 and 6 summarize the computation results of the example scenario. In these tables, we give the results of the example running using Fig. 7. For instance, in Table 5, we give the results of the path detection and threat computing procedures. We also determine the hop length of the identified paths. Likewise, Table 6 summarizes the focused points in terms of the number of hops, average threat value, and some maximum and minimum values, using the results in Table 5.

## Performance analysis

In this subsection, we present the simulation results and performance evaluations. To investigate the behaviors of proposed methods under various parameters, we change the number of nodes, the number of edges, the threat threshold, the hop threshold, and the hot spot threshold values. Therefore, the proposed algorithms run over various scalable network sizes, and we present a comprehensive performance analysis. The simulation results are presented considering average values of 1,000 iterations.

Figure 8 demonstrates the number of nodes to be removed against a varying number of nodes. In an IoT-based network graph, the hop threshold is set at 6, the threat threshold at 0.1, and the hot spot index at 3. According to Fig. 8, the number of removed paths decreases as the depending on the number of nodes increases. When new nodes are joined to the IoT-based network topology, the number of paths between the source and destination increases. The total threat value decreases when the path length and the total number of nodes are considered. This is because the desired metric is to reduce threat paths that have high total threat values and less hop count.

Figure 9 depicts the number of nodes to be removed against the number of edges. The hop threshold in an IoT-based network graph is set to 6, the threat threshold to 0.1, and the hot-spot index to 3. According to the figure, the number of nodes to be removed increases as the number of nodes increases. Since the number of paths between source and destination increases as the number of edges increases. Path length and the total number of edges are considered the overall threat value increases. If the cumulative threat value of the path exceeds the threat threshold value, the path is removed. Since the main goal of the procedure is to reduce threat paths with high cumulative threat values.

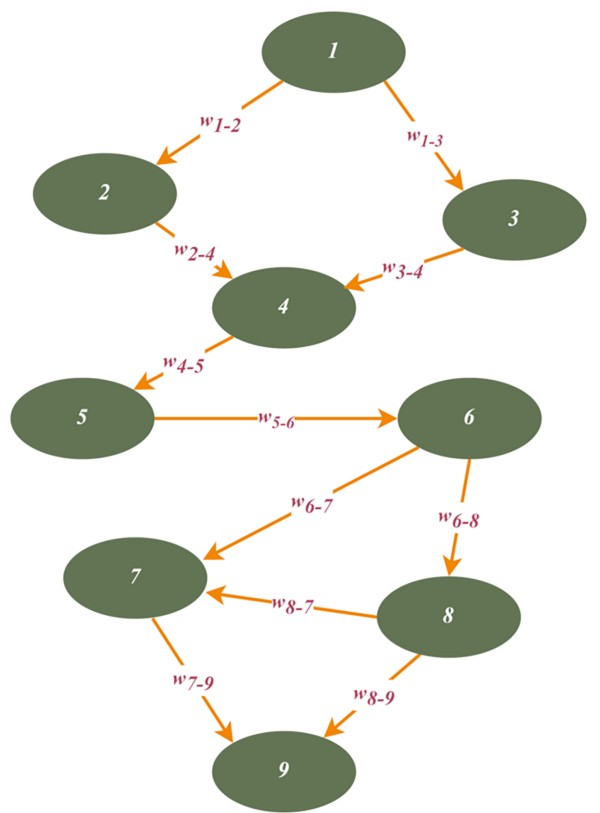

**Figure 7  An example of IoT-based network threat graph.**

**Table 5  General results of example scenario.**

| Path no | Path node sequence | Hop length | Avg threat |
|---|---|---|---|
| 1 | 1, 2, 4, 5, 6, 8, 7, 9 | 7 | 5.8 |
| 2 | 1, 3, 4, 5, 6, 8, 7, 9 | 7 | 6.6 |
| 3 | 1, 2, 4, 5, 6, 7, 9 | 6 | 5.05 |
| 4 | 1, 2, 4, 5, 6, 8, 9 | 6 | 5.85 |
| 5 | 1, 3, 4, 5, 6, 7, 9 | 6 | 5.83 |
| 6 | 1, 3, 4, 5, 6, 8, 9 | 6 | 5.8 |

**Table 6  Computation results of example scenario.**

| Computing metrics | Values | Path no |
|---|---|---|
| Maximum hop | 7 | 1, 2 |
| Minimum hop | 6 | 3, 4, 5, 6 |
| Average hop | 6.33 | – |
| Maximum threat | 6.6 | 2 |
| Minimum threat | 5.05 | 3 |
| Average threat | 5.82 | – |

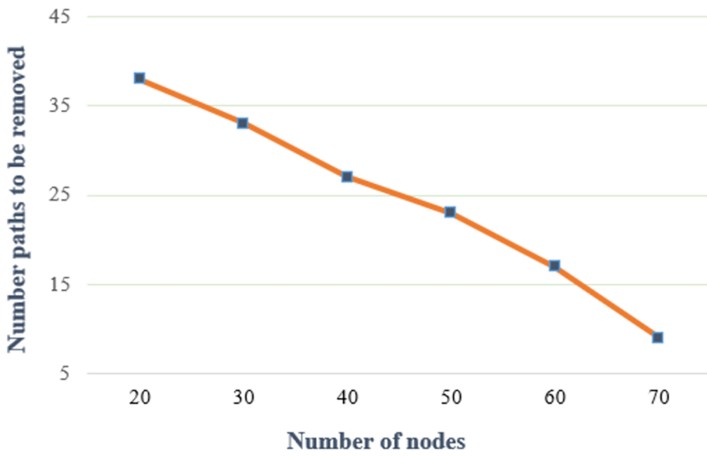

**Figure 8 Number of paths to be removed under various numbers of nodes.**

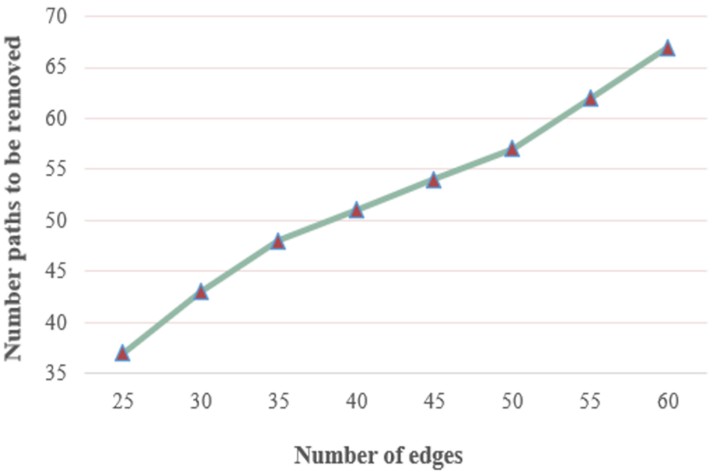

**Figure 9 Number of paths to be removed under various numbers of edges.**

Figure 10 demonstrates the number of nodes to be removed against varying hop thresholds. In an IoT-based network graph, the number of nodes is set at 20, the threat threshold at 0.1, and the hot-spot index at 3. According to the figure, the number of paths to be removed decreases as the hop threshold increases. Since, considering the fixed number of nodes, edges, and threat threshold, the number of paths from the source to the destination is constant. Accordingly, the total threat value and hop count of paths do not change. The increasing number of hops reduces the number of reduced paths due to the fixed path length.

Figure 11 shows the number of nodes to be removed under varying threat thresholds. The number of nodes is set at 20, the hop threshold at 6, and the hot-spot index at 3 for this running. It is clear that the number of paths to be removed decreases as the threat threshold increases. Since, the number of paths from the source to the destination is constant considering the fixed number of nodes, edges, and hop threshold. In addition, the

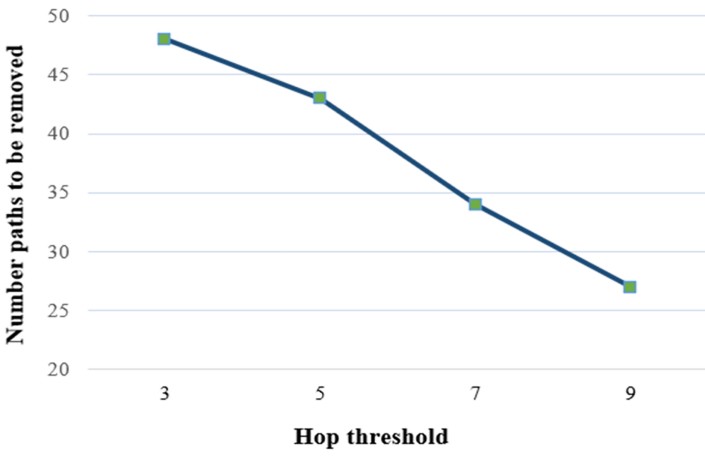

**Figure 10 Number of paths to be removed under various hop thresholds.**

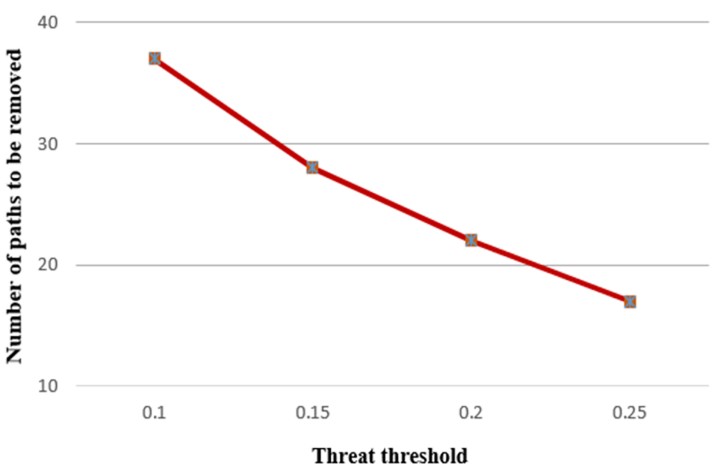

**Figure 11 Number of paths to be removed under various threat thresholds.**

total threat value and hop count of paths remain constant and do not change. The increase in the threat threshold values reduces the number of removed paths in the fixed path length condition.

Figure 12 demonstrates the running time results between *Arat & Akleylek (2023a)* and our studies in terms of threat path-reducing algorithms. According to the results, our proposed procedure performs better under an increased number of nodes. The path-reducing algorithm given in *Arat & Akleylek (2023a)* calls threat computing procedure and it works $\mathcal{O}(n^3)$ time complexity due to the Floyd-Warshall algorithm. Our proposed reducing procedure takes the threat amount as a parameter which is computed in the Algorithm 2, and it works linearly due to accessing the threat amount.

For the hot spot detection, according to Fig. 13, it is obvious that the number of hot spots detected for a fixed number of nodes decreases as the cut-off index increases. In addition, the number of detected hot spots decreases as the number of nodes increases for

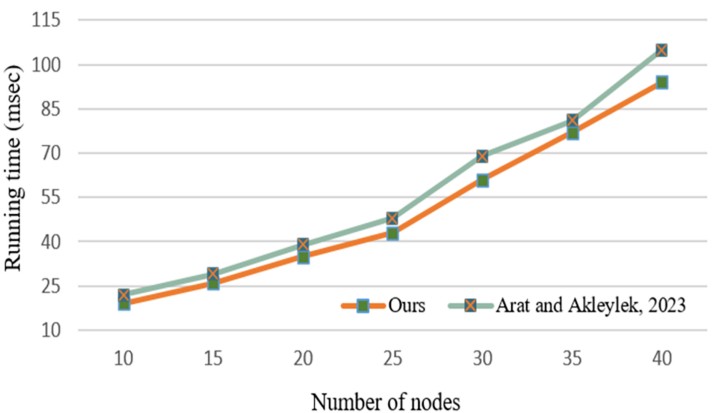

**Figure 12 Running time for reducing threat paths under various numbers of nodes.**

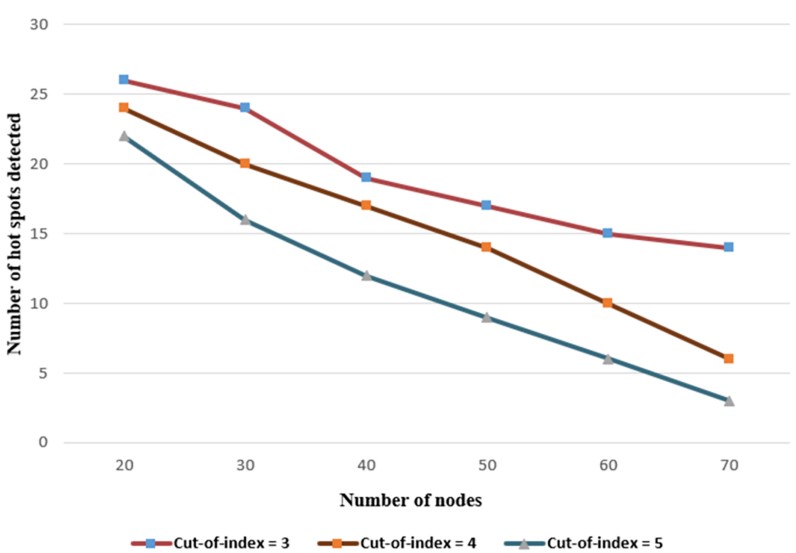

**Figure 13 Number of detected hot-spots under various numbers of nodes.**

the fixed cut-off index. The cut-off index determines the number of removed paths considering the hop threshold value.

Figure 14 highlights comparison results between *George & Thampi (2018)* and our studies in terms of running time due to threat path detection procedures. The comparison is made under various numbers of nodes and it is obvious that our proposed threat path detection procedure outperforms (*George & Thampi, 2018*). Since the proposed procedure works using the DFS algorithm, any additional variable is not used to store visited nodes on the traversed graph.

Figure 15 compares *George & Thampi (2018)* and our study in terms of running time due to hot spot finding procedures. As seen in the figure, the proposed hot spot detecting procedure outperforms (*George & Thampi, 2018*) under the varying number of nodes. The main difference between the studies is that our proposed procedure uses hash objects to

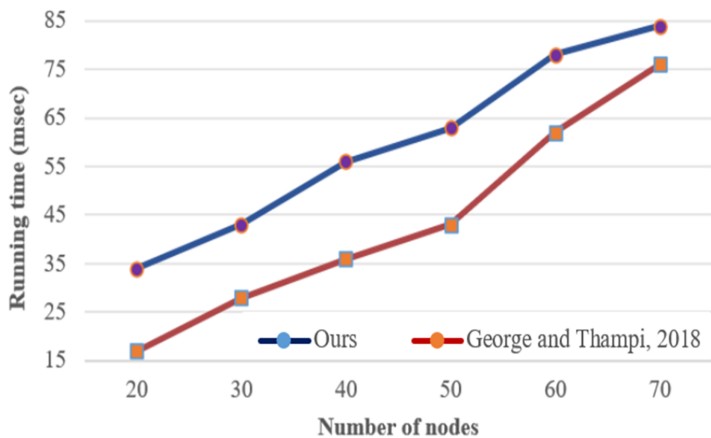

**Figure 14 Running time for detecting threat paths procedure under various number of nodes.**

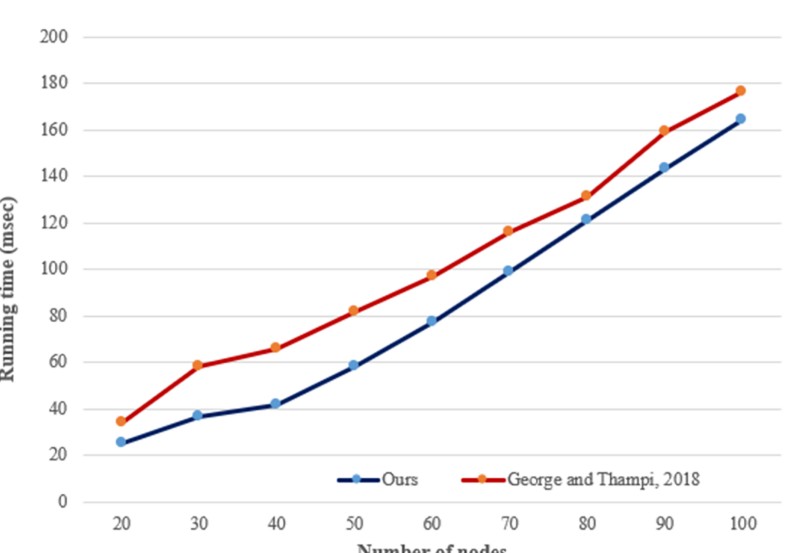

**Figure 15 Running time for finding hot spots under various numbers of nodes.**

store node data, labels, and hot spot values. Therefore, the procedure can access the stored data quickly. In addition, the proposed method performs linear and $\mathcal{O}(n)$ time complexity.

## CONCLUSION AND FUTURE WORKS

Detection and prevention of potential threat paths in network systems is a key issue for network and system administrators. Especially in IoT-based systems, which have high interaction and connections among devices, security vulnerability detection is essential. This study focuses on vulnerability and threat detection in IoT-based systems. Considering the existing graph-based assessment methods, threat paths are represented as a DAG. A weighted graph structure is generated using the CVSS base scores to help calculation of several parameters and metrics. In addition, the threat pathfinding method is modified using the DFS algorithm. All possible threat paths are detected by the DFS algorithm.

Therefore, the existing attack pathfinding procedure was improved in terms of the running time period. Using hop length and cumulative threat values, detected vulnerability paths are reduced considering the determined threshold values. As a final, a hot spot detection algorithm, which is used to compute node connections, is designed. Also, the hot spot detecting algorithm was improved in terms of running time complexity. The performance evaluation is made using a custom simulator, which is designed in the Python programming language. According to the evaluation results, the obtained performance outputs were presented. Proposed and modified algorithms provide detection and removal of threat paths on networks. Time complexities of existing threat paths and hot spot detecting procedures were reduced by proposed methods. We improved the previous study in terms of algorithmic approaches, complexity, and time efficiency. This means that we proposed compact and complete procedures in a different way in terms of pathfinding and threat computing. We also concatenated graph-reducing procedures according to a previous study. In future studies, attack tree modeling for other platforms such as avionics will be studied.

### Funding
This research was supported by ASELSAN A.Ş. The funders had no role in study design, data collection and analysis, decision to publish, or preparation of the manuscript.

### Grant Disclosures
The following grant information was disclosed by the authors:
ASELSAN A.Ş.

### Competing Interests
Sedat Akleylek is an Academic and Section Editor for PeerJ Computer Science.

### Author Contributions
- Ferhat Arat conceived and designed the experiments, performed the experiments, analyzed the data, performed the computation work, prepared figures and/or tables, authored or reviewed drafts of the article, and approved the final draft.
- Sedat Akleylek conceived and designed the experiments, performed the experiments, analyzed the data, performed the computation work, prepared figures and/or tables, authored or reviewed drafts of the article, and approved the final draft.

### Data Availability
The source codes are available in the Supplemental Files.

### Supplemental Information
Supplemental information for this article can be found online at http://dx.doi.org/10.7717/peerj-cs.1743#supplemental-information.

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
