# Peer review of "Modified graph-based algorithm to analyze security threats in IoT"

_PeerJ Computer Science, doi:10.7717/peerj-cs.1743_

## Round 0.1 · original submission · Major Revisions

Please be sure to address the reviewers' comments in your revised manuscript.

Reviewer 1 ·

Basic reporting

The paper is generally well-written. The authors use clear language throughout the paper. It includes several references to existing research, which helps readers understand the problem domain and its significance.

While the paper is generally self-contained, there are places where adding more details could make it easier to understand the material better. For example, in the section where the hop threshold, threat threshold, etc. are introduced, providing additional context may be helpful for those who may not have an in-depth understanding of this terminology. In the subsection which explains the proposed algorithm, the authors say: “The hash object hotSpotDict contains key and value pairs that are formed by nodes and hot-spot degrees”. While this provides a brief description of what the hash object contains, it might still be useful to clarify this for readers who may not be familiar with the term "hash object" and its role in the context of the research. A more detailed explanation could include things such as a brief description of what a "hash object" is and how it is used in this specific context, an explanation of what the "key and value pairs" represent within the hash object, and why the use of hash objects is important in this study and how it contributes to reducing running time, etc.

In the subsection “Representation of Threat Graph”, the authors introduce a Threat-Based Directed Graph (TDG) representation. Although this is illustrated using a specific example as shown in Figure 3, my suggestion is to provide a more generalized graphical representation. Such representation would give the reader a clearer, abstract visualization of the TDG, helping them to better understand the concept beyond any specific example.

Figures 3 to 6 represent different aspects of the research, specifically the TDG systems and their components. However, the descriptions of these figures are somewhat concise and lack a deeper connection to the overall narrative of the paper. Adding more details to these descriptions would enhance the technical quality of the manuscript. It would help readers understand how these descriptions relate to the bigger picture of the paper.

There are some issues with the readability of text within images. In figures 3-7, the text within the figure is quite small and difficult to read, which makes it difficult to understand the figure.

In the subsection “Performance Analysis”, when describing Figure 10 and Figure 11, it sounds like repeating the same thing, just with different numbers. My suggestion is to make it more concise by avoiding repetitive text.

Experimental design

The focus of this research is to address security challenges within IoT networks. The authors have introduced a modified graph-based approach to detect threats specific to such systems. They have identified a critical issue in the domain of IoT security and have discussed the significance of detecting and preventing potential threat paths in network systems. For this, they have introduced a custom-designed simulator using Python along with a simulation setup and scenario analysis. Such a setup has enabled the authors to assess and validate the proposed method across different scenarios. Since the proposed method relies on a simulation-based approach, providing more details on how their findings could be used in real-life situations would help readers understand the practical advantages of this research. Understanding how these methods might be applied in actual IoT networks would enhance the manuscript's overall contribution.

The paper provides a detailed description of the simulation setup and the creation of a threat-based graph. In “Example Scenario” under the Experiments section, the authors say: “In addition, the threat weights were set using CVSS metrics.”. Providing more details on how exactly CVSS metrics were translated into threat weights would give the reader a deeper understanding of this aspect. They also mentioned: “Table 5 and Table 6 summarize the computation results of the example scenario”. However, they haven’t thoroughly analyzed what these results mean or their significance. It would be valuable to know how these computations relate to their proposed security approach and how they contribute to addressing security challenges in IoT networks. Furthermore, including a brief discussion by the authors on ethical considerations such as data privacy and security would enhance the paper's overall quality.

The paper compares its method to George et al.'s approach. But it's not clear why only this method was chosen for comparison and not others. Explaining this choice would help readers understand why this specific comparison matters.

Validity of the findings

The authors have used a custom-designed simulator to simulate and analyze threat paths in IoT-based network systems. They have emphasized the importance of reducing time complexities compared to existing procedures, underlining the need for efficient threat detection methods.
However, in evaluating the validity of the findings, it is important to investigate further into the technical aspect of the proposed approach.

The explanation provided by the authors, stating that their proposed procedure uses hash objects to store node data, labels, and hot spot values for quick data access, is somewhat limited. “…our proposed procedure uses hash objects to store node data, labels, and hot spot values. Therefore, the procedure can access the stored data quickly.” While this statement highlights a key distinction between their approach and others, it lacks sufficient detail to fully understand the significance of this difference. A more elaborate explanation of how the use of hash objects improves data access efficiency and why this matters in the context of their research would enhance the clarity and depth of their argument. Providing specific examples or comparisons with alternative data storage methods would further strengthen this aspect of the paper.

The explanations related to Figures 14 and 15, comparing their approach with George et al.'s method, lack depth and comprehensiveness. While the figures show the running time for threat path detection and hot spot identification, the text provides minimal context or analysis regarding these comparisons. The authors simply say “...our proposed threat path detection procedure outperforms (George and Thampi, 2018) and “...the proposed hot spot detecting procedure outperforms (George and Thampi, 2018) under the varying number of nodes” without providing any reasonable explanation. The authors should provide a more detailed discussion, highlighting how their approach compares to the referenced method, the implications of these comparisons, and why these differences matter. Addressing these aspects would enhance the research's technical rigor and would enhance its contribution to the domain of IoT security.

Cite this review as

Reviewer 2 ·

Basic reporting

There are only a limited number of papers that discuss attack representation in IoT and IIoT networks. In addition, very few studies focus on threat path reductions, considering threat path weights and path lengths; therefore, they are proposing a new graphical structure for representing threats inside a network.

The problem is relevant today because as the utilization of IoT networks increases, attackers often choose new ways to find their attack paths, but in this paper, they have considered the threat path reduction method as a comparison with George and Thampi (2018), and therefore only a few parameters are considered for reducing such paths, but there are a lot more to be addressed for threat path reduction.

Experimental design

The proposed method uses a DFS approach to find the paths toward the target, which is a common method to find the paths in the attack graph. Furthermore, considering attack paths, and weighted links, the number of paths is reduced using hop count, but as compared with George and Thampi (2018), the complexity of the algorithms is reduced.

When they have reduced the path {src−i−t −m−dest}, but there are still {src-k-m-dest} and {src-k-r-dest} which are the lowest hop counts to be removed. In the next step, they detected the hot spots and removed the path {src-k-r-dest} and why not {src-k-m-dest}. So it is still confusing whether they are trying to remove or keep the critical attack paths. It will be better if a good use case for an IoT or IIoT network is chosen and explained well.

More study regarding CVE is required, In the paper, they have mentioned CVE as a program, but it is a database or dictionary of known vulnerabilities.

Overall, the writing style can be improved. References are adequate.

Validity of the findings

A new graph-based approach to represent general threat types in IoT-based networks, but how this proposed graph is different from other existing graphical structures is not mentioned properly. The proper definition of a threat and the relationship between threats to form a graphical structure are not clearly discussed.

Cite this review as

---

## Round 0.2 · Minor Revisions

There are some minor revisions that need to be addressed.

Reviewer 1 ·

Basic reporting

Overall, the paper is well-written. It does require some work to improve its overall quality, particularly in terms of clarity, structure, grammar, and readability. No major issues were identified.

There's a repetition in the section 'Path Reduction With Threat Threshold': "In addition, it will be easier for system administrators and system administrators…".

Experimental design

The authors have sufficiently addressed the concerns previously raised with regard to the experimental design.

Validity of the findings

Adequate revisions have been made by the authors.

Cite this review as

Reviewer 2 ·

Basic reporting

The authors have incorporated all of my suggestions, and the manuscript can be accepted for publication.

Experimental design

The authors have incorporated all of my suggestions, and the manuscript can be accepted for publication.

Validity of the findings

The authors have incorporated all of my suggestions, and the manuscript can be accepted for publication.

Cite this review as

---

## Round 0.3 · accepted · Accept

The authors have addressed the remaining concerns. The paper is ready to be published.